# Prohibitin 2 is Involved in Parkin-Mediated Mitophagy in Urothelial Cells of Cattle Infected with Bovine Papillomavirus

**DOI:** 10.3390/pathogens9080621

**Published:** 2020-07-29

**Authors:** Francesca De Falco, Ivan Gentile, Pellegrino Cerino, Anna Cutarelli, Cornel Catoi, Sante Roperto

**Affiliations:** 1Dipartimento di Medicina Veterinaria e Produzioni Animali, Università degli Studi di Napoli Federico II, 80137 Napoli, Italia; francesca.defalco@unina.it; 2Dipartimento di Medicina Clinica e Chirurgia, Università degli Studi di Napoli Federico II, 80131 Napoli, Italia; ivan.gentile@unina.it; 3Istituto Zooprofilattico Sperimentale del Mezzogiorno, 80055 Napoli, Italia; strategia@izsmportici.it (P.C.); annacutarelli@hotmail.it (A.C.); 4Pathology Department, Faculty of Veterinary Medicine, University of Agricultural Sciences and Veterinary Medicine, 400374 Cluj-Napoca, Romania; cornel.catoi@usamvcluj.ro

**Keywords:** Bovine papillomavirus, ERAS, optic atrophy 1 (OPA1), parkin-dependent mitophagy, prohibitin-2 (PHB2), transcription factor EB (TFEB)

## Abstract

Prohibitin 2 (PHB2), an inner mitochondrial membrane (IMM) protein, has recently been identified as a novel receptor involved in parkin-mediated mitophagy. In the field of veterinary medicine, the role of PHB2 in parkin-mediated mitophagy was described, for the first time, in urothelial cells of cattle, naturally infected with bovine papillomavirus (BPV). The BPV2 and BPV13 E5 oncoprotein, responsible for abortive infections in urothelial cells, was detected by RT-PCR. Severe ultrastructural abnormalities of the inner mitochondrial membrane were detected using transmission electron microscopy. PHB2 formed a functional complex with PHB1. PHB2 was significantly overexpressed in mitochondrial fractions from urothelial mucosa samples taken from cattle harbouring BPV infection. PHB2 overexpression could be attributed to mitochondrial dysfunction, as its expression levels in the cytosolic, microsomal, and nuclear fractions were seen to be unmodified. Immunoprecipitation studies revealed the interaction between PHB2 and phosphorylated forms of both PINK1 and parkin. Furthermore, PHB2 interacted with LC3-II, a marker of autophagosomal membranes and autophagy receptors, such as p62 and optineurin. PHB2 was shown to interact with transcription factor EB (TFEB), which is activated following parkin-mediated mitophagy, and embryonic stem cell-expressed Ras (ERAS), a constitutive protein coded by ERas. Western blot analysis revealed a significant overexpression of unphosphorylated TFEB in mitochondrial and nuclear fractions from urothelial mucosa samples from cattle suffering from BPV infection. Finally, PHB2 interacted with ERAS, believed to be involved in mitophagosome maturation. Taken together, the molecular and ultrastructural findings of this study suggested that BPV infection is responsible for parkin-dependent mitophagy, in the pathway of which PHB2 plays a crucial role.

## 1. Introduction

Mitochondria are double membrane organelles, consisting of an outer mitochondrial membrane (OMM) surrounding an inner mitochondrial membrane (IMM) [1]. Mitochondria are regulated by constant fusion and fission events, referred to as mitochondrial dynamics. Damaged and dysfunctional mitochondria are removed by selective autophagy, termed mitophagy, that is an evolutionarily conserved cellular process [2,3]. Mitophagy, the sequestration of mitochondria by autophagosomes followed by their degradation in lysosomes, contributes to maintaining the quality of the mitochondrial population, thus mediating cell survival and viability in response to infection [4,5,6]. Mitophagy can be mediated by the E3 ubiquitin ligase parkin that acts cooperatively with the Ser/Thr kinase PINK1. Mitophagy has also been found to be mediated by protein receptors that interact with LC3 through their LC3-interacting region (LIR). Receptors such as NIX/BNIP3L, BNIP3, and FUNDC1 have been identified as mitophagy receptors in mammalian cells [7]. More recently, novel mitophagy receptors have been found. They comprise prohibitin 2 (PHB2), localised in the IMM [8] and FKBP8, a member of the FK506-binding protein family, anchored in the OMM [9]; both the mitophagy receptors have an LIR. 

PHBs are evolutionarily conserved from bacteria to humans and are a ubiquitously expressed family of membrane proteins. Phylogenetic analysis revealed that PHBs in all species can be divided into two types, namely PHB1 and PHB2, which together form a high molecular weight complex in the IMM [10,11]. PHB2 appears to play a central role in parkin-mediated mitophagy [8,12,13]. Indeed, PHB2 promotes PINK1/parkin-dependent mitophagy through the PARL-PGAM5-PINK1 axis [14]. It has been shown that PHB2-mediated mitophagy attenuates injury in renal tubular epithelial cells by regulating mitochondrial dysfunction [15]. Furthermore, it has been suggested that PHB2 is required for cholestasis-induced mitophagy, wherein PHB2 recruits LC3 to the damaged mitochondria via interaction with p62/SQSTM1 and LC3 [16]. Recently, PHB2 overexpression has been shown to be associated with aggressiveness in diffuse large B-cell lymphomas and targeting PHBs may have a therapeutic potential, notably in the aggressive subtypes [17]. PHBs are believed to be play a crucial a role in the pathogenesis of virus infection. Indeed, PHBs contribute to retain virus on cell surface and enhance intracellular virus replication [18,19].

Bovine papillomaviruses (BPVs) comprised 27 members grouped into five genera (https://pave.niaid.nih.gov/). Recently, the genetic characterisation of an additional, putative BPV belonging to Xipapillomavirus has been done [20]. BPV-2 and BPV-13 belong to Deltapapillomavirus (δPV) and are the most significant infectious agents associated with bladder tumours in some breeds of pasture-residing cattle that graze on bracken fern-infested lands [21,22,23]. BPV-2 and BPV-13 have been shown to cause both abortive and productive infections resulting in neoplastic and non-neoplastic pathology. 

δPV E5 is believed to be the major oncoprotein of δPVs [21]; it has a small size, comprising 40-85 amino acids, and is the most studied protein, which displays transforming activity via numerous pathways even in the absence of other viral genes.

The present study aims to report the molecular and ultrastructural characteristics of the IMM protein, PHB2, as a key mitophagy receptor for parkin-mediated mitophagy in bovine PV-infected urothelial cells. 

## 2. Materials and Methods

### 2.1. Ethics Statement

In this study we did not perform any animal experiments. All samples were collected post-mortem from slaughterhouses and no ethics approval was required.

### 2.2. Tumour Samples

Neoplastic urothelial samples from 15 3- to 25-year-old cows clinically suffering from chronic enzootic haematuria were collected from public slaughterhouses after bladder neoplasms had been identified during routine meat inspection. Additionally, normal mucosa samples of the urinary bladder were collected from 10 3- to 16-year-old healthy cows. Both groups were composed of animals that belonged to the Podolica breed, which graze on lands rich in bracken fern. Neoplastic and healthy bladder mucosa samples were subdivided and either fixed in 10% buffered formalin for subsequent microscopy-based experiments or immediately frozen in liquid nitrogen and stored at −80 °C for molecular biology analysis. 

### 2.3. Transmission Electron Microscope (TEM)

Urothelial mucosa samples from cattle suffering from naturally occurring bladder cancer were fixed in 4% glutaraldehyde in 0.1 M phosphate buffer (pH 7.4) for 2–3 h, and postfixed in 1% osmium tetroxide (OsO_4_) in the same buffer for 1 h. Thereafter, the samples were washed in 0.1 M phosphate buffer (pH 7.4), dehydrated in graded alcohol and embedded in Agar Low Viscosity Resin (AGR 1078) (Agar Scientific Limited, Essex, England). Urothelial mucosa samples from healthy cattle were processed similarly. Semi-thin sections (400 nm) were cut with a glass knife on an EM UC6 ultramicrotome (Microsystems CMS GmbH-DCC, Wetzlar, Germany) and stained with 1% aqueous toluidine blue. Ultrathin sections (60–70 nm), obtained using the same ultramicrotome with a diamond knife, were collected onto 300-mesh copper grids coated with formvar and counterstained with lead citrate and UranyLess, an aqueous solution from Electron Microscopy Sciences (Hatfield, PA, USA). The sections were observed with a JEOL JEM-1011 transmission electron microscope (JEOL, Tokyo, Japan), equipped with a thermionic tungsten filament and operated at an acceleration voltage of 100 kV. Images were taken using a Morada cooled slow-scan CCD camera (3783 × 2672 pixels) and micrographs were taken with iTEM software (Olympus Soft Imaging System GmbH, Munster, Germany). 

### 2.4. Antibodies

Rabbit antibodies against embryonic stem cell-expressed Ras (ERAS) and phospho-parkin (Ser65) were obtained from Biorbyt (St. Louis, MO, USA). Rabbit antibodies against PHB1, optic atrophy 1 (OPA1), LC3, p62, total transcription factor EB (TFEB), phospho-TFEB (Ser211), optineurin (OPTN), PHB1, Tom 20, Calnexin, and HSP90 were obtained from Cell Signaling (Leiden, Netherlands). Rabbit anti-PINK1, goat anti- parkin, and mouse anti-PHB2, c-Myc and β-actin were from Santa Cruz Biotechnology (Dallas, TX, USA). Sheep anti-phospho-PINK1 (Thr257) was purchased from Ubiquigent (Dundee, SC, UK). 

### 2.5. RNA Extraction and Reverse Transcription (RT)-PCR

Total RNA was extracted from 15 bovine urothelial tumour samples and 10 bladder samples from healthy cows by the RNeasy Mini Kit (Qiagen, NW, DE), according to the manufacturer’s instructions. Genomic DNA was removed from the RNA preparations using the RNase-free DNase Fermentas Life Sciences (Thermo Fisher Scientific, Waltham, MA, USA). A total of 1 µg of RNA was used to generate single stranded cDNA, using the QuantiTect Reverse Transcription Kit (Qiagen NW, DE), according to the manufacturer’s instructions. PCR was performed with a specific primer set designed using Primer3, an online tool, for the BPV-2 and BPV-13 E5 gene, and bovine PHB2. The following primers were used: BPV-2 E5 ORF forward 5′-CACTGCCATTTGTTTTTTTC-3′, reverse 5′-GGAGCACTCAAAATGATCCC-3′; BPV-13 E5 ORF forward 5′-CACTGCCATTTGGTGTTCTT-3′, reverse 5′- AGCAGTCAAAATGATCCCAA-3′; PHB2 forward 5’-GCTCCAAAGACCTGCAAATGG-3’; reverse 5’-AGCACCTCGTTGACAATGGA-3’. Conditions for PCR were: 94 °C for 5 min, 40 cycles at 95 °C for 30 s, 58 °C for 30 s, and 72 °C for 30 s.

### 2.6. Sequence Analysis

PCR products, obtained by RT-PCR, were purified using the Qiaquick PCR purification Kit (Qiagen NW, DE) and bidirectionally sequenced using the Big Dye-Terminator v1.1 Cycle Sequencing Kit (Applied Biosystems, CA, USA), as per the manufacturer’s recommendations. Sequences were dye terminator removed by DyeEx-2.0 spin kit (Qiagen) and run on a SeqStudio Genetic Analyzer (Thermo Fischer Scientific, Waltham, MA, USA). Electropherograms were analysed using Sequencing Analysis v5.2 and Sequence Scanner v1.0 software (Thermo Fischer Scientific, Waltham, MA, USA). The sequences were analysed using BLAST.

### 2.7. Real-time RT-PCR

To perform real-time RT-PCR analysis, total RNA and cDNA from neoplastic and healthy urinary bladder samples were generated, as described above. Real-time PCR was carried out with a Bio-Rad CFX Connect™ Real Time PCR Detection System (Bio-Rad, Hercules, CA, USA), using iTAq Universal SYBR® Green Supermix (Bio-Rad). Each reaction was done in triplicate and the primers used for PHB2 were the same as that for the RT-PCR. The thermal profile for the PCR was 95 °C for 10 min, 40 cycles of 94 °C for 15 s, and 58 °C for 30 s, followed by a melting curve. Relative quantification (RQ) was calculated by using the CFX Manager™ software, based on the equation RQ = 2 − ΔΔCq, where Cq is the quantification cycle to detect fluorescence. Cq data were normalised to the bovine β-actin gene (forward: 5′-TAGCACAGGCCTCTCGCCTTCGT-3′, reverse5′-GCACATGCCGGAGCCGTTGT-3′).

### 2.8. Western Blot Analysis

Healthy and neoplastic bovine urothelial samples were lysed in radioimmunoprecipitation assay (RIPA) buffer (50 mM Tris-HCl, (pH 7.5), 1% Triton X-100, 400 mM NaCl, and 1 mM ethylenediaminetetraacetic acid), 2 mM phenylmethylsulfonyl fluoride, 1.7 mg/mL aprotinin, 50 mM NaF, and 1 mM sodium orthovanadate. Protein concentration was measured using the Bradford assay (Bio-Rad). For Western blotting, 50 μg of protein lysate was heated at 90 °C in 4X premixed Laemmli sample buffer (Bio-Rad), clarified by centrifugation, separated by sodium dodecyl sulphate–polyacrylamide gel electrophoresis, and transferred onto nitrocellulose membranes (GE Healthcare, UK). Membranes were blocked with Tris-buffered saline and 0.1% Tween 20, containing 5% non-fat dry milk for 1 h at room temperature. The membranes were subsequently incubated overnight at 4 °C with primary antibodies, washed thrice with TBST, incubated for 1 h at room temperature with goat anti-rabbit or goat anti-mouse (Bio-Rad) or donkey anti-goat or rabbit anti-sheep (Santa Cruz Biotechnology) HRP conjugated secondary antibodies, diluted at 1:5000 in TBST, and washed thrice with TBST. Immunoreactive bands were detected using the Western Blotting Luminol Reagent (Santa Cruz Biotechnology) and the ChemiDoc XRS Plus (Bio-Rad). Images were acquired with Image Lab Software version 2.0.1.

### 2.9. Cell Fractionation 

Mitochondrial, cytosolic, microsomal and nuclear fractions from urothelial tumour and healthy cells were prepared using the Qproteome Mitochondria Isolation Kit (Qiagen, NW, DE). Briefly, 60 mg tissue from each sample was washed in 0.9% NaCl and incubated for 10 min at 4 °C in lysis buffer. The homogenate was centrifuged at 1000 g for 10 min at 4 °C; the supernatant was designated as the cytosolic fraction. The pellet was resuspended in disruption buffer and passed through a 26-gauge needle 15 times. The fraction enriched in nuclei was pelleted by centrifugation at 1000 g for 10 min and was homogenised in disruption buffer. To obtain the enriched mitochondrial fraction, the supernatant was centrifuged at 6000 g for 10 min at 4 °C. The pellet contained mitochondria and the supernatant constituted the microsomal fraction. The pellet was resuspended in mitochondria storage buffer. All buffers, except the mitochondria storage buffer, were supplemented with protease inhibitors at 1:100, provided with the kit. Protein concentration of different fractions was determined with the Bio-Rad protein assay kit (Bio-Rad). Tom 20 as mitochondrial, Calnexin as microsomal, HSP90 as cytosolic and c-Myc as nuclear markers were utilised.

### 2.10. Mitochondrial, Cytosolic and Nuclear Immunoprecipitation

Mitochondrial, cytosolic, and nuclear extracts from normal and pathological bladders, obtained as previously described, were immunoprecipitated. Protein samples (100 μg) were incubated with anti-PHB2 antibody or anti-mouse IgG (isotype) and with anti-PHB1 antibody or anti-rabbit IgG (Bethyl Laboratories, Inc., Montgomery, TX, USA) for 1 h at 4 °C with gentle shaking. Thereafter, the samples were centrifuged at 1000 *g*, for 5 min at 4 °C and incubated with 30 μL of Protein A/G-Plus Agarose (sc-2003) (Santa Cruz Biotechnology) overnight at 4 °C. The immunoprecipitates were washed four times in complete lysis buffer and separated on polyacrylamide gels. Subsequently, the proteins were transferred onto nitrocellulose membranes. The membranes were blocked for 1 h at room temperature in 5% bovine serum albumin, and then incubated with primary antibodies overnight at 4 °C. After three washes in TBS, the membranes were incubated with secondary antibodies for 1 h at room temperature. Chemiluminescent signals were developed with the Western Blotting Luminol Reagent (Santa Cruz Biotechnology) and detected using the ChemiDoc XRS gel documentation system (Bio-Rad).

### 2.11. Statistical Analysis

Results are presented as means ± standard error (SE). Data were assessed by one-way analysis of variance (ANOVA), followed by the Tukey’s test for multiple comparisons of means using the GraphPad PRISM software version 8 (GraphPad Software, San Diego, CA, USA). A *p*-value ≤ 0.05 indicated statistical significance.

## 3. Results

### 3.1. Virological and Ultrastructural Findings

We have been investigating microscopic and submicroscopic patterns of urothelial samples from cattle clinically suffering from chronic enzootic haematuria that have lived free at pasture and grazed on lands rich in bracken fern. This clinical syndrome is prevalently caused by bladder tumours that are very commonly associated with BPV infection [24,25]. Therefore, we performed virological investigation to validate our hypothesis that papillomavirus infection, detected by E5 ORF expression required to bladder carcinogenesis, could occur in neoplastic bladder mucosa samples. Indeed, virological findings of all samples were characterised by the expression of BPV-2 and BPV-13 E5 ORF, as RT-PCR revealed 154 and 153 bp E5 transcripts, respectively (Figure 1). E5 transcripts were not detected in non-neoplastic bladder mucosa samples.

Mitochondrial abnormalities were seen at the ultrastructural level. Invaginations of the IMM protruding into the mitochondrial matrix were commonly seen. Some of these invaginations detached from the IMM, forming vesicles within the electron lucent matrix (Figure 2). 

Aberrant fragmentation of mitochondrial cristae was another common ultrastructural finding. Furthermore, some cristae were characterised by an abnormal width of the cristae junction (CJ) and the cristae lumen (CL), with a vesicle-like appearance. Double-membraned structures, similar to isolation membranes or phagophores, appeared to be localised to damaged mitochondria at focal rupture sites of the OMM (Appendix A).

### 3.2. Proteins Responsible for Mitochondrial Integrity: OPA1 and the PHB Complex

OPA1 and the PHB complex play a crucial role in mitochondrial integrity. OPA1 mediates IMM fusion and preserves cristae morphogenesis [26]. Therefore, we investigated its expression. Western blot analysis performed on total extracts revealed an evident OPA1 expression both in healthy and diseased samples (Figure 3).

Western blot analysis was also performed on fractionated extracts and an OPA1 overexpression was found at the mitochondrial level only (Figure 4A), where a *p* value ≤ 0.05 showed statistical significance (Figure 4B), which suggested that a change in OPA1 levels could be involved in cristae fragmentation. Notably, overexpression of OPA1 has been associated with mitochondrial fragmentation [27]. 

OPA1 expression is known to be regulated by the PHB complex, which, thereby, controls cristae morphogenesis and the functional integrity of mitochondria [28]. Therefore, we investigated the expression of PHB1 and PHB2. We detected PHB1, by Western blot analysis, in immunoprecipitates using an anti-PHB2 antibody (Figure 8), and PHB2 in immunoprecipitates using an anti-PHB1 antibody (Figure 5), suggesting that PHB1 and PHB2 form a functional complex in the urothelial cells of cattle.

### 3.3. PHB2 Expression Profile

PHB2 is believed to be a key receptor of parkin-dependent mitophagy [8]. Therefore, we have focused our investigation on the expression profile of PHB2. RT-PCR investigation of PHB2 cDNA obtained from healthy and diseased bladder mucosa samples detected a 139-bp transcript, the alignment of which showed 100% identity with bovine mRNA PHB2 sequences deposited in GenBank (accession number NM_001046198) (Appendix A). 

Western blot analysis performed on total extracts revealed a statistically significant PHB2 overexpression (*p* ≤ 0.05) in urothelial cells of bladder mucosa from BPV-infected cattle in comparison with urothelial cells from healthy cattle (Figure 6A,B). 

Notably, immunoblotting performed on subcellular fractions revealed a significant (*p* ≤ 0.05) overexpression of mitochondrial PHB2 only in urothelial cells infected by BPVs in comparison with non-infected urothelial cells. Unmodified PHB2 expression levels were seen at the cytosolic and nuclear levels of both healthy and pathological urothelial cells (Figure 7A,B). 

Furthermore, we performed real-time PCR on PHB2 cDNA and found no statistically significant difference in the transcript levels between diseased and healthy urothelial cells (Appendix A), which may suggest, according to Signorile et al. [29], that overexpression of PHB2 is not due to upregulation of gene expression but due to mitochondrial dysfunction.

### 3.4. The IMM Resident PHB2 Interacts with the Phosphorylated PINK1/Parkin Axis

It has been suggested that PINK1, a mitochondrial kinase, undergoes autophosphorylation to become a highly active functional kinase (pPINK1), which is a prerequisite for the mitochondrial translocation of parkin [30]. pPINK1 phosphorylates parkin at Ser65 (pParkin) [31]. The phosphorylation of PINK1/parkin is an essential prerequisite for the activation of parkin-mediated mitophagy [32], and PHB2 is the key receptor required for parkin-mediated mitophagy in cultured cells [8].

Western blot analysis detected the presence of pPINK1 and p-parkin in anti-PHB2 mitochondrial and cytosolic immunoprecipitates in both healthy and pathological bladder mucosa samples harbouring a spontaneous BPV infection, suggesting that PHB2 interacts with the functional PINK1/parkin axis (Figure 8).

Furthermore, parkin-mediated mitophagy has been shown to be upregulated in urothelial cells of cattle infected by BPVs [33]. Experimental studies showed that PHB2 is essential for parkin-mediated mitophagy [8]. Consistent with molecular findings from experimental studies, it is conceivable that PHB2 is a mitochondrial receptor for parkin-mediated mitophagy in this spontaneous model of viral infection. Therefore, our study suggests that PHB2 is a mitochondrial receptor for parkin-mediated mitophagy in a spontaneous model of viral infection.

### 3.5. PHB2 Binds to LC3-II and Autophagy Receptors (p62 and OPTN)

Although both PHB1 and PHB2 have an LC3-interacting region (LIR), only PHB2 has been shown to bind directly to the autophagosome-associated protein LC3. To validate this previously unreported interaction in the urothelial cells of cattle, we performed Western blot analysis on anti-PHB2 immunoprecipitates of subcellular fractions (cytosol and mitochondria) and showed the presence of LC3 in both diseased and healthy samples. However, in the mitochondrial fractions of papillomavirus-harbouring bladder mucosa samples, LC3-II levels were appreciably higher than in mitochondrial fractions from healthy bladder mucosa samples (Figure 8), suggesting activation of mitophagy, as LC3-II, the lipidated component of LC3 protein, plays a role in the recognition of damaged mitochondria, destined for degradation [34]. It has been suggested that OMM rupture is necessary for the binding of PHB2 to LC3-II [8]. Ultrastructural findings from our study seem to support this hypothesis. Moreover, we detected p62 in anti-PHB2 mitochondrial immunoprecipitates which appeared to be expressed less in mitochondrial fractions from cells infected by bovine papillomavirus than mitochondrial fractions from healthy bladder cells (Figure 8). This corroborates previous studies indicating that: (a) mitophagy is activated in cells harbouring BPV infection as p62 is selectively degraded by autophagy [35]; (b) p62 may be essential for parkin-mediated mitophagy in urothelial cells of cattle spontaneously infected by BPV, similar to that seen in cultured cells [30]. 

OPTN, a novel autophagy receptor, is recruited, independently on p62, to damaged mitochondria for regulating their autophagic turnover during parkin-mediated mitophagy [36]. OPTN interacts with LC3 through its LIR; its potential interplay, synergism, and functional redundancy with other autophagy receptors have been suggested [37]. Therefore, we investigated the expression profile of this autophagy receptor. Western blot analysis detected an OPTN expression both in healthy and neoplastic samples (Figure 9). 

Immunoblotting performed on subcellular fractions revealed a significant (*p* ≤ 0.05) overexpression of mitochondrial OPTN only in urothelial cells infected by BPVs in comparison with non-infected urothelial cells. Unchanged OPTN expression levels were seen at the cytosolic and microsomal levels of both healthy and pathological urothelial cells (Figure 10A,B). 

This suggests that BPV infection is responsible for the activation of mitophagy, as OPTN overexpression is known to be directly involved in autophagosome formation [38]. Finally, in both the cytosolic and the mitochondrial anti-PHB2 immunoprecipitates, we detected the presence of OPTN (Figure 8), which corroborates experimental data showing that PHB2 and OPTN play a role in parkin-dependent mitophagy. 

### 3.6. PHB2 Interacts with TFEB and Eras

In the present study, we report a remarkable increase in peculiar ultrastructural feature in mitochondria and lysosomes of urothelial cells harbouring BPV. In cultured cells, it has been shown that mitochondrial and lysosome biogenesis is activated by transcription factor EB (TFEB) upon induction of parkin-mediated mitophagy [39]. Therefore, we performed Western blot analysis to better understand the TFEB expression profile as TFEB is a master regulator of autophagy and its overexpression results in the clearance of damaged mitochondria [40,41]. We detected a significant reduction of expression levels of phosphorylated TFEB (pTFEB) in neoplastic samples in comparison with healthy samples (Figure 11A,B). 

Western blot analysis was also performed to detect expression of unphosphorylated, active TFEB. We detected evident expression levels of TFEB on total extracts both from healthy and neoplastic samples (Figure 12).

Using cellular subfractionation, significant increased expression levels (*p* ≤ 0.05) of total TFEB were seen in both the nucleus and mitochondria in samples from neoplastic mucosa in comparison with healthy mucosa (Figure 13A,B), suggesting that nuclear translocation of active TFEB occurred in urothelial cells spontaneously infected with BPVs. 

It has been suggested that PHB is a novel dynamic regulator of mitochondrial and nuclear function in cultured cells as an anti-apoptotic molecule and a transcriptional regulator [42]. We hypothesised that PHB2 may be involved in the nuclear translocation of active, total TFEB. To validate this hypothesis, we performed an immunoprecipitation study, using an anti-PHB2 antibody, on subcellular fractions. Western blot analysis performed on these immunoprecipitates revealed that expression levels of total TFEB were appreciably higher in the nuclear fractions of BPV-infected urothelial cells than in urothelial cells from healthy cattle (Figure 14), which suggested that PHB2 may play a role in shuttling TFEB to the nucleus from the cytoplasm. 

It has been shown that, unlike in humans, Eras is a functional gene in adult cattle [43]. Furthermore, ERAS has been shown to be overexpressed in urothelial cells infected by BPVs, a functional partner of the BPV E5 oncoprotein [44]. Recently, ERAS was shown to be involved in mitophagic machinery [33,45]. Therefore, we wanted to investigate whether ERAS could be involved in molecular signalling of mitophagy mediated by PHB2. Co-immunoprecipitation, using an anti-PHB2 antibody, detected the presence of ERAS in both the mitochondrial and the cytosolic fractions both in healthy and BPV-harbouring urothelial cells (Figure 8), which showed that ERAS physically interacted with PHB2. Furthermore, PHB2 co-precipitated ERAS more appreciably in cells from bladder mucosa infected by BPVs rather than in cells from healthy mucosa, which suggested that this functional stochiometric complex was upregulated in activated mitophagy upon BPV infection.

## 4. Discussion

Mitophagy is activated in urothelial cells infected by BPVs [33,45,46]. The present study identified, for the first time in comparative medicine, that the IMM protein PHB2 is a receptor involved in parkin-mediated mitophagy in urothelial cells from cattle suffering from a spontaneous infection by BPVs. Our data demonstrated that PHB2 constitutes a conserved mitophagy receptor that operates at the IMM and, after being unmasked upon outer membrane permeabilisation, facilitates the autophagic removal of the organelle. A few studies have investigated the role of PHB2 as a receptor for mitophagy and most of them have been performed in vitro [14]. To date, very few studies have investigated the in vivo role of PHB2 as the IMM mitophagy receptor [16]. 

Our data showed that in anti-PHB2 immunoprecipitates, pPINK1 and p-parkin were more abundant in virus-infected mucosa samples than in healthy mucosa samples. As the phosphorylation of both PINK1 and parkin is crucial for selective autophagy of damaged mitochondria [47], it is conceivable that PHB2 is involved in parkin-mediated mitophagy in urothelial cells harbouring BPV infection, and our recent study demonstrated that BPV-E5 oncoprotein may upregulate parkin-mediated mitophagy [33]. Our results from this spontaneous model of disease corroborate previous in vitro studies showing that PHB2 plays a crucial role in parkin-dependent mitophagy [8,14]. Our results showed that PHB2 bound to the autophagosomal membrane-associated protein LC3-II. Furthermore, PHB2 was found to bind to p62, which, in turn, appeared to be significantly reduced in cells harbouring BPV infection in comparison with healthy cells. Notably, p62 is selectively degraded by autophagy [35]. A significant increase of OPTN was also detected in anti-PHB2 immunoprecipitates from pathological samples. It is conceivable that OPTN could play a role in enhancing LC3-II production to promote autophagosome formation. Furthermore, it may coordinate the autophagosome-mediated engulfment of cargo as well as vesicular trafficking and amplify parkin-mediated mitophagy via its LIR moiety. Our hypothesis appears to be supported by recent in vitro experiments [37,38,48]. Prohibitin-dependent OPA1 stabilisation is required for controlling mitochondrial architecture and dynamics [49,50,51]. Indeed, PHB2 and OPA1 regulate inner membrane integrity and have essential roles in regulating cristae morphogenesis and structure [28,52]. PHBs and OPA1 regulate the mitochondrial membrane potential and, ultimately, oxidative phosphorylation (OXPHOS) [53]. It has been shown that abnormal mitochondrial cristae are responsible for bioenergetic dysfunction, since cristae function as bioenergetic units [53]. Mammalian cells with abnormal OXPHOS due to aberrant cristae show mitochondrial fragmentation [5,54]. Furthermore, it has been suggested that bioenergetic dysfunction and/or impaired cristae formation may result in vesicle formation in the mitochondrial matrix [53,55]. It is conceivable that OPA1 overexpression may be responsible for cristae fragmentation thus contributing to alterations in mitochondrial architecture that is critical in promoting parkin-dependent mitophagy in urothelial cells spontaneously infected by bovine papillomavirus. It is also conceivable that OPA1 overexpression may play a role in the biogenesis of matrix vesicles, as seen in this study. Our assumption is consistent with experimental data which suggested that OPA1 controls the width of the CJs and CLs [56] and that OPA1 overexpression triggers mitochondrial fragmentation [27], thus upregulating parkin-mediated mitophagy [57]. OPA1 overexpression is known to confer apoptotic resistance and increase the susceptibility of cells towards selective autophagy [26,54,58], whereas the loss of OPA1 increases apoptotic sensitivity of cells [27,59]. 

Surprisingly, co-immunoprecipitation studies revealed that PHB2 interacted with TFEB and ERAS. In vitro, it has been shown that the nuclear translocation of TFEB occurs during mitophagy [40]. This is the first time that PHB2 has been shown to interact with TFEB in vivo. It is conceivable that this interaction may facilitate TFEB translocation to the nucleus, which would explain the statistically significant increase of total, active nuclear TFEB expression of our study. Indeed, experimental studies suggested that PHBs shuttle between the mitochondria and the nucleus as a transcriptional regulator [42]. Nuclear TFEB is known to be a master regulator of the lysosomal system [60,61] as the cellular localisation and activity of TFEB are strictly controlled by its phosphorylation status. Phosphorylated TFEB remains inactive in the cytoplasm, whereas total, unphosphorylated TFEB translocates to the nucleus and activates the transcription of its target genes [41]. Therefore, TFEB could coordinate a transcriptional program to control cellular degradative pathways by increasing the number of lysosomes and the level of lysosomal enzymes, in a spontaneous model of BPV infection. PHB2 could also contribute to mitochondrial biogenesis by promoting nuclear translocation of active TFEB in urothelial cells upon parkin-mediated mitophagy, induced by BPV infection. Our assumption is consistent with experimental data showing translocation of the TFEB to the nucleus upon induction of parkin-dependent mitophagy, which results in mitochondrial and lysosomal biogenesis, accompanied with an increase in lysosomal enzyme levels [39]. Therefore, the functional TFEB/PHB2 complex could represent an interaction by which PHB2 plays a crucial role as a transcriptional regulator through a novel molecular pathway based on TFEB signalling. 

ERAS is a constitutive GTPase in cells of adult cattle, and is typically localised to the plasma membrane [43]. ERAS shares several common functions with the small GTPase, Rheb. Indeed, ERAS and Rheb are involved in the phosphatidylinositol 3-kinase (PI3K) pathway [62]. Recently, Rheb has been shown to promote mitophagy after being recruited to the mitochondria [63]. Furthermore, ERAS is known to interact with the phosphorylated platelet derived growth factor β receptor (pPDGFβR), a key partner of BPV E5 oncoprotein in the urothelial cells of cattle [44]. It has been suggested that ERAS contributes to the activation of the PI3K/Akt pathway in BPV infection [44]. Recently, it has been shown that ERAS plays a role in mitophagy and is upregulated by the BPV E5 oncoprotein. ERAS interacted with BNIP3/Nix mitophagy receptors in parkin-independent mitophagy [45] as well as with the PINK1/parkin axis in parkin-dependent mitophagy [33]. It is conceivable that ERAS plays a role in autophagosome maturation, including mitophagosome maturation, via the PI3K signalling pathway, as it has been shown that phosphatidylinositol 3-phosphate (PI3P) plays a crucial role in the expansion of autophagosomal membranes until closure [64]. 

## 5. Conclusions

There is a scant information concerning the role(s) of selective autophagy, including mitophagy, in health and disease of domestic animals. Although mitochondria are emerging as an essential signalling hub in regulation of both innate and adaptive immunity in eukaryotic cells and are becoming an attractive target for developing new therapies, our knowledge of mitophagy during viral infection is in the initial stage in comparative medicine. Our study focuses on role of PHB2 as a novel mitophagy receptor in urothelial cells of cattle harbouring BPV infection. There are few studies on this topic, thereby, mechanisms of the IMM PHB2 remain to be elucidated in details [12]. It will be interesting to investigate if and how PHB2 interacts with other component of the mitophagy machinery such as FUNDC1 and BNIP3/Nix receptors. Very likely, PHB2 and other prohibitins will represent the topics of specific interest in next future as further details in PHB2 biology will contribute to gain insights into unexplored mitophagy mechanisms and virus infections. A further exciting research area will be to highlight PHB2 as an important target which may be useful in diagnosis and therapy of PHB-dependent diseases, including viral diseases [15,17].

## Figures and Tables

**Figure 1 pathogens-09-00621-f001:**
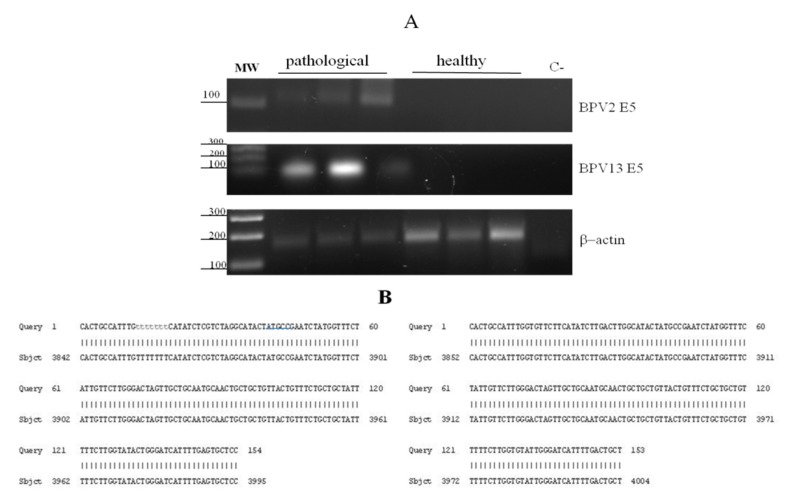
RT-PCR and amplicon sequences for BPV-2 and BPV-13 E5. (**A**) Electrophoresis of RT-PCR products for the analysis of BPV-2 and BPV-13 E5 mRNA expression in healthy and pathological bovine bladders. Lane MW: DNA molecular weight marker (100-base pair ladder); lanes 2–4: three representative papillary urothelial cancer samples; lanes 5–7: healthy bladder samples; lane C-: no template control (no cDNA added). (**B**) Amplicon sequences showing 100% identity with BPV-2 and BPV-13 E5 sequences deposited in GenBank (accession number: M20219.1 and JQ798171.1, respectively).

**Figure 2 pathogens-09-00621-f002:**
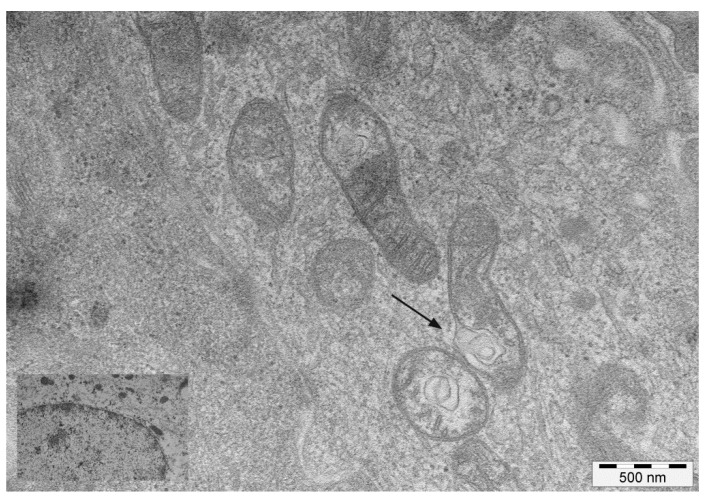
TEM images of mitochondria. Transmission electron microscopy images of mitochondria showing invaginations of the inner membrane (arrow), detachment of which results in mitochondria-derived vesicles in the matrix. Inset: mitochondria in healthy urothelial cells.

**Figure 3 pathogens-09-00621-f003:**
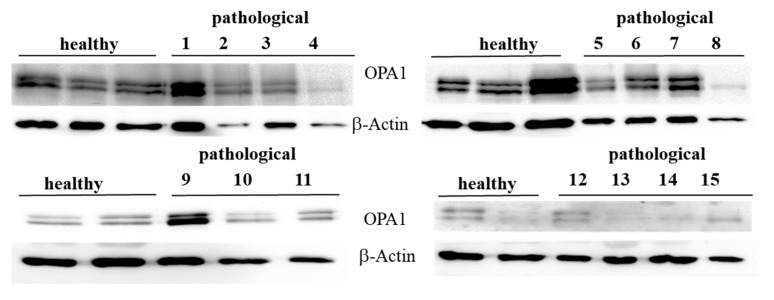
Western blot analysis of OPA1 protein. Western blot analysis of OPA1 protein from total extracts performed on 10 healthy and 15 neoplastic bladder mucosa samples.

**Figure 4 pathogens-09-00621-f004:**
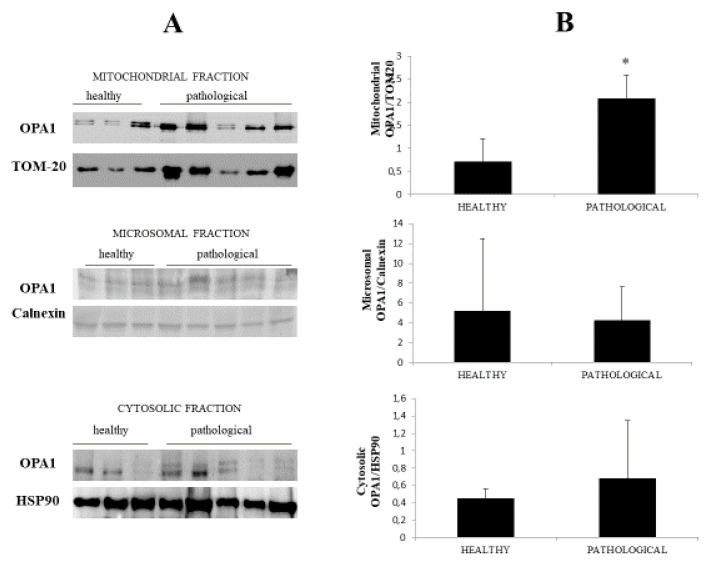
Western blot analysis of OPA1. (**A**) Western blot analysis of OPA1 subcellular fractions performed on all healthy and neoplastic bladder mucosa samples. Representative samples from three healthy mucosa and five neoplastic urothelium (three high-grade and two low-grade carcinomas) are shown. (**B**) Densitometric analysis obtained by comparing the expression levels of mitochondrial OPA1 with TOM 20 showed that OPA1 was significantly overexpressed in the neoplastic bladder mucosa samples (* *p* ≤ 0.05).

**Figure 5 pathogens-09-00621-f005:**
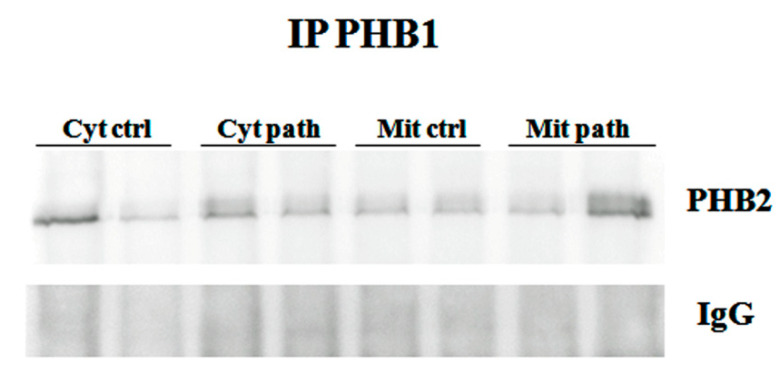
IP PHB1. Mitochondrial and cytosolic immunoprecipitation using an anti-PHB1 antibody in non-neoplastic and neoplastic bladder samples. Western blot analysis detected PHB2, which suggested that PHB1 and PHB2 formed a functional complex. IgG: control isotype. Cyt crtl and Cyt path = cytosolic fraction, control and pathological; Mit crtl and Mit path = mitocondrial fraction, control and pathological).

**Figure 6 pathogens-09-00621-f006:**
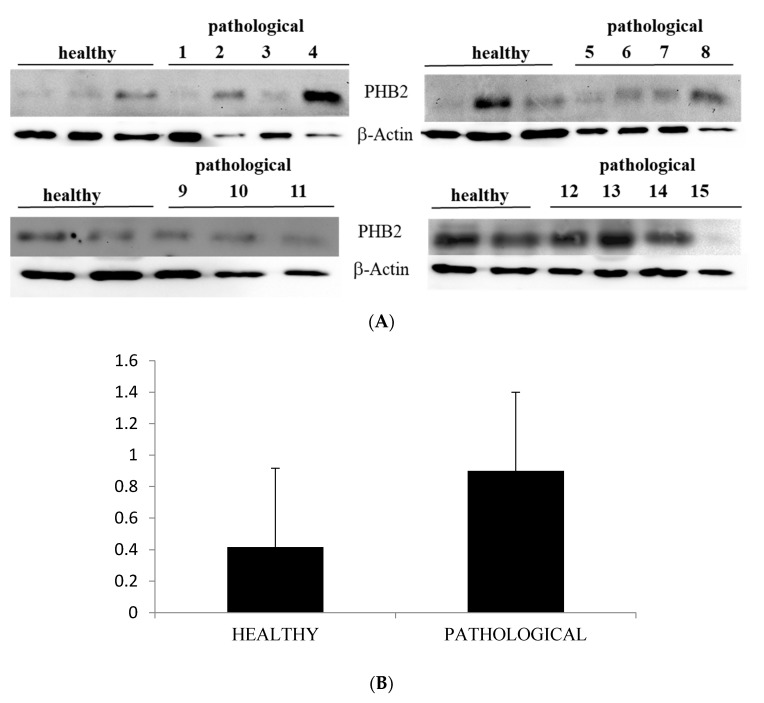
Western blot analysis of PHB2 protein. (**A**) Western blot analysis of PHB2 protein from total extracts in 10 healthy and 15 neoplastic samples. (**B**) Densitometric analysis of PHB2 protein relative to β-actin protein levels: PHB2 protein level was overexpressed in neoplastic samples in a statistically significant manner. The calculations were based on three independent determinations. The values for the latter are expressed as a percentage of the average values for the healthy samples (* *p* ≤ 0.05).

**Figure 7 pathogens-09-00621-f007:**
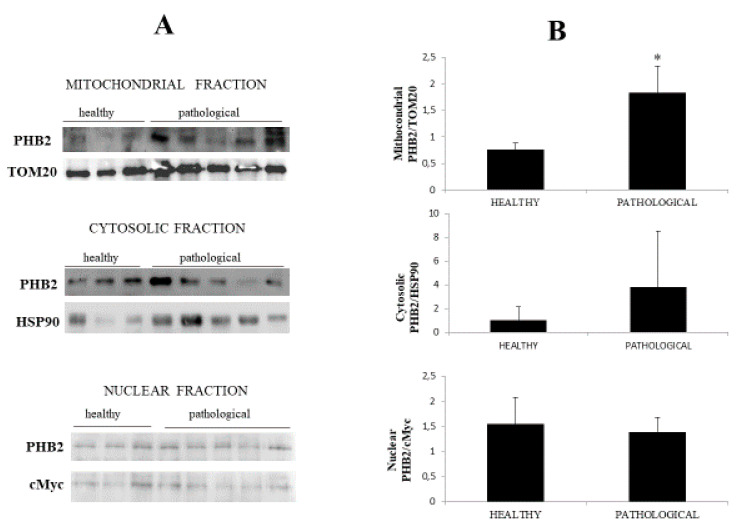
Western blot analysis of PHB2. (**A**) Western blot analysis of PHB2 in normal and neoplastic bovine bladder samples in the subcellular fractions. (**B**) Densitometric analysis of mitochondrial PHB2 level was performed relative to Tom 20 protein level. HSP90 and c-Myc were used as cytosolic and nuclear markers, respectively. PHB2 was found to be significantly overexpressed at the mitochondrial level in samples from neoplastic bladder mucosa. The calculations were based on three independent determinations. The values for the latter are expressed as a percentage of the average values for the healthy samples (* *p* ≤ 0.05).

**Figure 8 pathogens-09-00621-f008:**
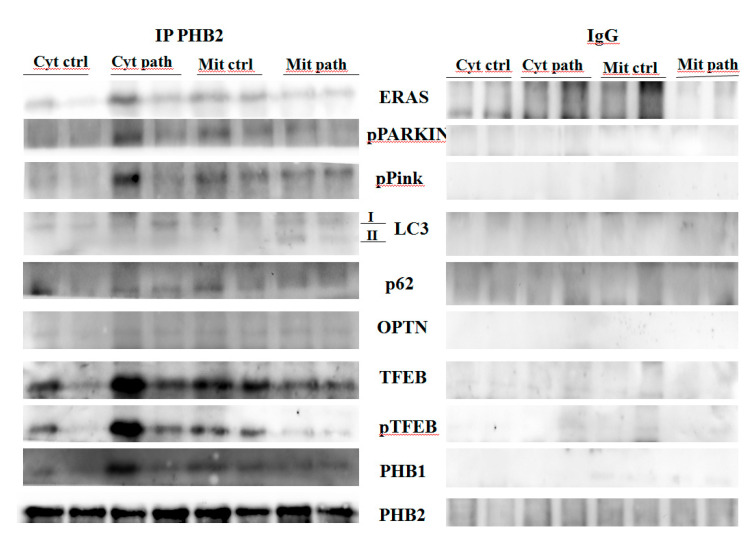
IP PHB2. Mitochondrial and cytosolic immunoprecipitation using an anti-PHB2 antibody in non-neoplastic and neoplastic bladder samples. Western blot analysis showing that PHB2 interacts with ERAS, pParkin, pPINK1, LC3, p62, OPTN, TFEB, pTFEB and PHB1. IgG: control isotype. Cyt crtl and Cyt path = Cytosolic fraction, control and pathological; Mit crtl and Mit path = mitochondrial fraction, control and pathological).

**Figure 9 pathogens-09-00621-f009:**
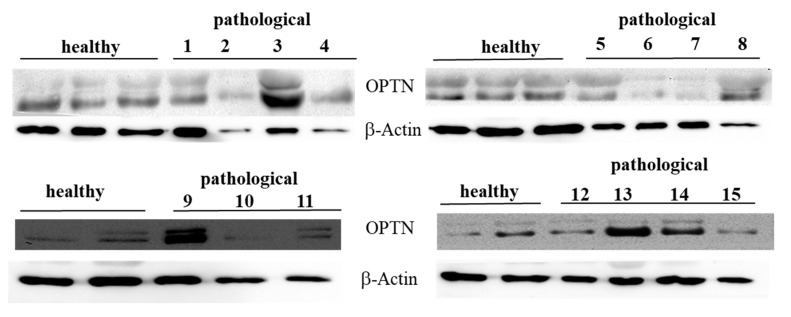
Western blot analysis of total OPTN protein. Western blot analysis of total OPTN protein performed on 10 healthy and 15 neoplastic bladder mucosa samples.

**Figure 10 pathogens-09-00621-f010:**
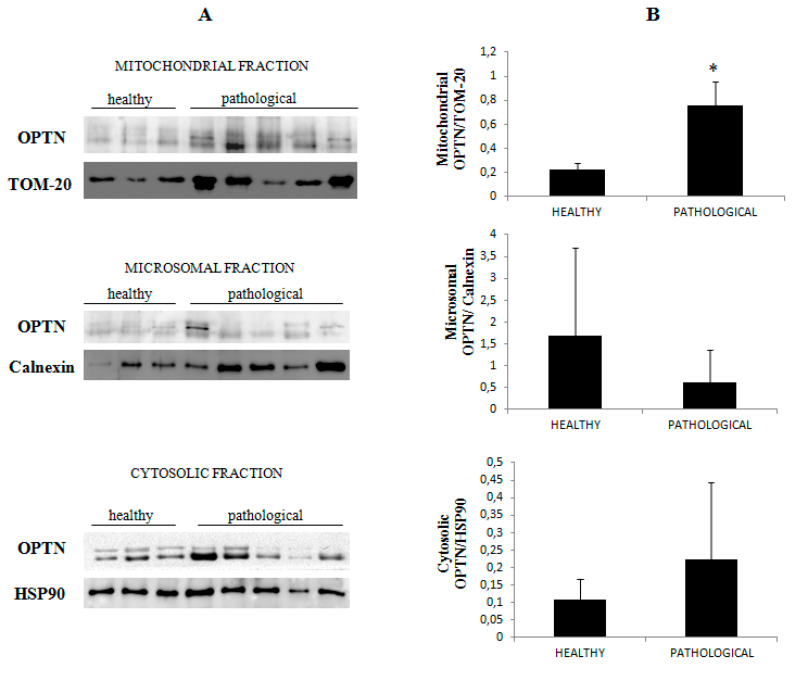
Western blot analysis of OPTN. (**A**) Western blot analysis of OPTN in normal and neoplastic bovine bladder samples in the subcellular fractions. (**B**) Densitometric analysis of mitochondrial OPTN level was performed relative to Tom 20 protein level. Calnexin and HSP90 were used as microsomal and cytosolic markers. OPTN was found to be significantly overexpressed at the mitochondrial level in samples from neoplastic bladder mucosa (* *p* ≤ 0.05).

**Figure 11 pathogens-09-00621-f011:**
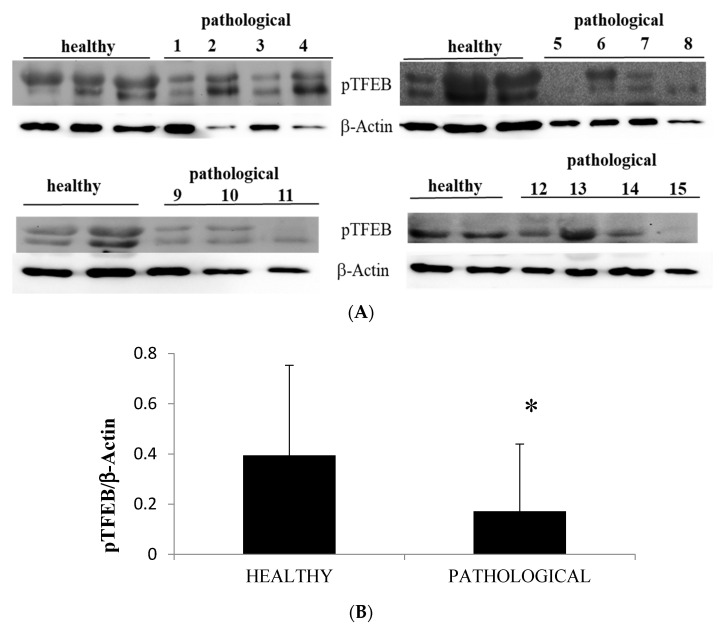
Western blot analysis of pTFEB. (**A**) Western blot analysis of phosphorylated TFEB (pTFEB) in the total lysate in healthy and neoplastic samples. (**B**) Densitometric analysis of pTFEB protein was performed in comparison with β-actin protein levels. The calculations were based on three independent determinations. The values for the latter are expressed as percentages of the average values for the healthy samples (* *p* ≤ 0.05).

**Figure 12 pathogens-09-00621-f012:**
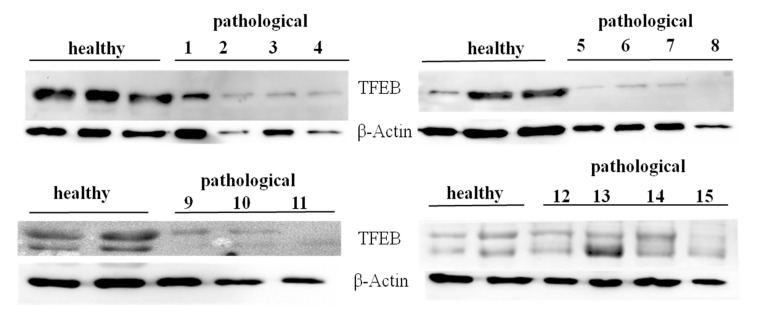
Western blot analysis of total TFEB protein. Western blot analysis of total, unphosphorylated TFEB protein performed on 10 healthy and 15 neoplastic bladder mucosa samples.

**Figure 13 pathogens-09-00621-f013:**
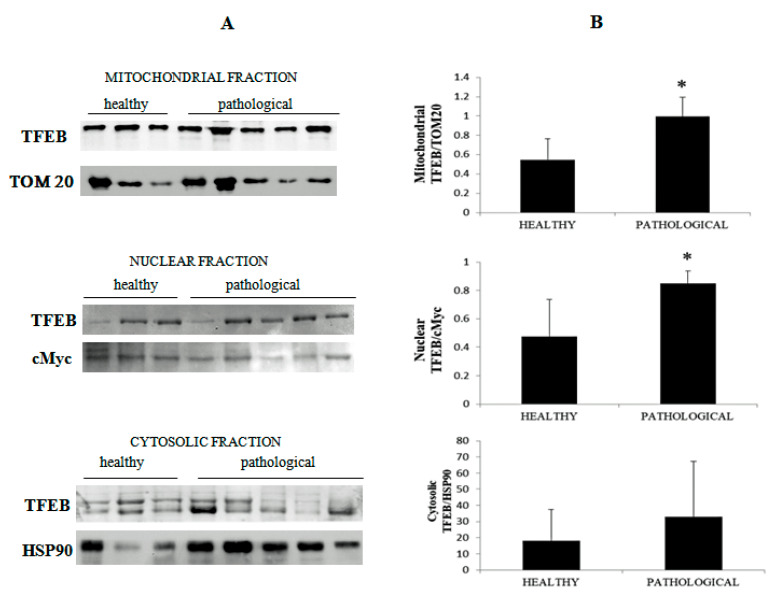
Western blot analysis of TFEB. (**A**) Western blot analysis of total TFEB in the subcellular fractions of healthy and neoplastic samples. (**B**) Densitometric analysis for TFEB proteins was performed comparing the expression levels of total TFEB with Tom 20 as mitochondrial, c-Myc as nuclear and HSP90 as cytosolic markers, respectively, and showed that mitochondrial and nuclear total TFEB was significantly overexpressed in the neoplastic bladder mucosa samples (* *p* ≤ 0.05).

**Figure 14 pathogens-09-00621-f014:**
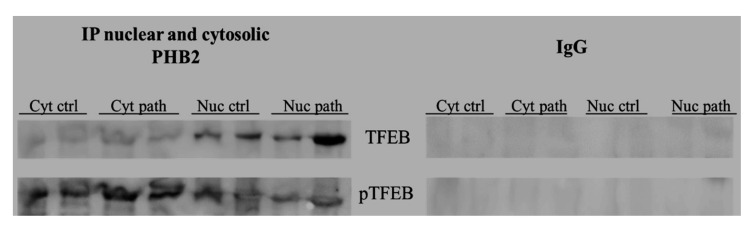
IP PHB2. PHB2 interaction with TFEB detected by Western blot analysis performed on cytosolic and nuclear extracts immunoprecipitated using an anti-PHB2 antibody in non-neoplastic and neoplastic bladder samples. IgG: control isotype. Cyt crtl and path = Cytosolic fraction, control and pathological; Nuc crtl and path = Nuclear fraction, control and pathological).

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
