# Peer review of "Prohibitin 2 is Involved in Parkin-Mediated Mitophagy in Urothelial Cells of Cattle Infected with Bovine Papillomavirus"

_pathogens, 2020, doi:10.3390/pathogens9080621_

Round 1

Reviewer 1 Report

The quality of the revised manuscript is greatly improved. 

Author Response

Thank you very much for your comments.

Reviewer 2 Report

In this article, authors showed that several protein essentials for mitochondrial quality control such as OPA1, OPTN and prohibin (PHB)2 are over-represented especially at mitochondrial fraction of BPV infected neoplastic urothelial cells of cattle. They hypothesized that stability of mitochondrial PHB2 is increased due to mitochondrial dysfunction in urothelial smple with BPV. Their previous study reported that BPV E5 upregulates parkin-dependent mitophagy, therefore further analysis was performed to examine possible interaction between PHB2 and other proteins known to play important roles in mitophagy was examined mainly by co-immunoprecipitation assays. They found that PHB1, PINK, parkin, OPA1, OPTN and LC3 all co-immunoprecipitated with PHB2 more efficiently in neoplastic BPV-1 positive samples than in normal urothelial samples. They concluded that BPV infection is responsible for increased parkin-mediated mitophagy and PHB2 plays a critical role in this BPV-1 induced mitophagy. I am not sure what are significant new findings in this article. Although authors claims that PHB2 acting as receptor of the parkin mediated mitophagy is new, this particular pathway is previously demonstrated in other experimental setting by other group as cited by the authors (Wei et al., Cell, 2017). It is not clear what is a take home message here.

Author Response

In this article, authors showed that several protein essentials for mitochondrial quality control such as OPA1, OPTN and prohibin (PHB)2 are over-represented especially at mitochondrial fraction of BPV infected neoplastic urothelial cells of cattle. They hypothesized that stability of mitochondrial PHB2 is increased due to mitochondrial dysfunction in urothelial smple with BPV. Their previous study reported that BPV E5 upregulates parkin-dependent mitophagy, therefore further analysis was performed to examine possible interaction between PHB2 and other proteins known to play important roles in mitophagy was examined mainly by co-immunoprecipitation assays. They found that PHB1, PINK, parkin, OPA1, OPTN and LC3 all co-immunoprecipitated with PHB2 more efficiently in neoplastic BPV-1 positive samples than in normal urothelial samples. They concluded that BPV infection is responsible for increased parkin-mediated mitophagy and PHB2 plays a critical role in this BPV-1 induced mitophagy. I am not sure what are significant new findings in this article. Although authors claims that PHB2 acting as receptor of the parkin mediated mitophagy is new, this particular pathway is previously demonstrated in other experimental setting by other group as cited by the authors (Wei et al., Cell, 2017). It is not clear what is a take home message here.

   In 2017, Wei et al. described, for the first time, prohibitin 2 (PHB2) as an inner mitochondrial membrane (IMM) mitophagy receptor. The authors performed their experimental study using mammalian cells such as murine embrionic fibroblasts (MEFs) and HeLa cells. To the best of our knowledges, since then a few papers have been published describing PHB2 as mitophagy receptors. In addition, all of these papers, that, anyway, we report in the References of our manuscript, deal with experimental studies except for Xiao et al’s (Cell Death and Disease 9, 160, 2018) and Zhang et al’s (Med Sci Monit 26, e923227, 2020). The latter studies were partly performed in vivo reporting PHB2 mitophagy in 65 biliary atresia patients and in 38 persons suffering from non-small cell lung carcininomas (NSCLCs), respectively.

We believe our study may be of interest for the following reasons:

  1. There are not in vivo studies describing PHB2 as mitophagy receptor in comparative virology so far; furthermore, just afew months ago Xu et al. (Autophagy 16, 3-17, 2020) have written;….further studies are required to investigate the complex relationship between PHB, mitochondrial dysfunction and mitophagy…. which shows that the role of PHB2 in mitohagy remains unclear.
  2. Our study is the first reporting potential role of PHB2 in mitophagy induced by viral infection in a spontaneous disease model;
  3. Our study reports for the first time PHB2 as mitophagy receptor in veterinary medicine;
  4. Our study wants to be a contribution to better understand the role of PHB2 in mitophagy keeping in mind that, as suggested also by He et al. (Acta Biochim Biophys Sin 49, 750-75, 2017), it remains uncertain about the properties of IMM protein PHB2 during mitophagy; however, according to Galluzzi et al (Curr Biol 27, R57-R76, 2017), we surmise that PHB2 will be the center of renewed interest over the forthcoming years.

Round 2

Reviewer 2 Report

Authors has made compelling arguments that this article is worth considering for publication. I agree that the topic is interesting and author’s finding have high potential for new discovery. On the other hand, their interpretation for some of results should be made more carefully and fairly. It seems a little eager to jump to conclusions.

Specific points are followings.

1) At first and foremost, there is no conclusive evidence that PHB2 is indeed working as a mitophagy receptor. What is shown is that PHB2 interacts with pPINK and parkin. Just because PHB2 interacts with parkin, it does not necessary means its working as a receptor.

As authors indicated, Wei et al reported that PHB2 is a mitophagy receptor and their previous study showed that parkin mediated mitophagy is increased in urothelial cells of cattle with BPV infection. I understand it is highly possible that PHB2 is indeed working as a receptor. However, without any further evidence, for example whether PHB2 knockdown abrogates parkin mediated mitophagy, their results are not sufficient to claim that “out study confirms that PHB2….(line 334 p12)”.

2) They claimed that LC3-II level l in PHB2 co-immunoprecipitates are higher in BPV positive samples than in negative healthy samples. However, total PHB2 level in the immunoprecipitates was not indicated. To evaluate the interaction was enhanced, it is important to compare ratio of the target protein per PHB2 between in BPV negative VS positive samples. They showed that PHB2 level is higher in BPV positive samples in Figure 6. Then, its possible that more PHB2 in the immunoprecipiates make LC3-II appears increased even when the level of interaction remains similar .

3) It is very difficult to see if p62 level in PHB2 co-immunoprecipitates was less in mitochondrial fraction of BPV positive samples than in that of BPV negative samples in Figure 8 (indicated as figure 7) due to weak detection. In discussion, authors claimed that its significantly reduced (line 453 p17). More clear results need to be shown to support such discussion.

4) In addition to 3), is there any result to support that the recruitment of p62 is dependent on functional parkin in this article (line 353 p13). For example, perhaps inhibiting parkin?

5) In figure 12, it seems less total TFEB in pathological samples than it is in healthy samples, yet its higher in pathological samples after fractionation shown in Figure 13. why?

6) As for ERAS interaction with PHB2 (line 435 p17), total PHB2 in the immunoprecipiates needs to be indicated. The same arguments here as 2) and 3),

7) Throughout the article, authors described that neoplastic samples are infected with BPV or harboring BPV. However, they only showed 3 samples express BPV E5 in figure S2. I believe not 100% of the disease is BPV positive. Thus, they should indicate how many samples are BPV positive and whether all “pathological sample” examined in this particle are BPV positive more clearly.

Minor points:

8) In line 343p12 and line 350 p13, its figure 8 instead of figure 7.

Author Response

We would like to thank this Reviewer for her/his comments allowing us to improve our manuscript.

Reviewer: 2

Comments and Suggestions for Authors

Authors has made compelling arguments that this article is worth considering for publication. I agree that the topic is interesting and author’s finding have high potential for new discovery. On the other hand, their interpretation for some of results should be made more carefully and fairly. It seems a little eager to jump to conclusions.

Specific points are followings.

1) At first and foremost, there is no conclusive evidence that PHB2 is indeed working as a mitophagy receptor. What is shown is that PHB2 interacts with pPINK and parkin. Just because PHB2 interacts with parkin, it does not necessary means its working as a receptor.

As authors indicated, Wei et al reported that PHB2 is a mitophagy receptor and their previous study showed that parkin mediated mitophagy is increased in urothelial cells of cattle with BPV infection. I understand it is highly possible that PHB2 is indeed working as a receptor. However, without any further evidence, for example whether PHB2 knockdown abrogates parkin mediated mitophagy, their results are not sufficient to claim that “out study confirms that PHB2…(line 334 p12)”.

   We completely agree with this very useful comment, thereby we modified the sentence (lines 336-339):

Experimental studies showed that PHB2 is essential for parkin-mediated mitophagy [8]. Consistent with molecular findings from experimental studies, it is conceivable that PHB2 is a mitochondrial receptor for parkin-mediated mitophagy in this spontaneous model of viral infection.

2) They claimed that LC3-II level l in PHB2 co-immunoprecipitates are higher in BPV positive samples than in negative healthy samples. However, total PHB2 level in the immunoprecipitates was not indicated. To evaluate the interaction was enhanced, it is important to compare ratio of the target protein per PHB2 between in BPV negative VS positive samples. They showed that PHB2 level is higher in BPV positive samples in Figure 6. Then, its possible that more PHB2 in the immunoprecipiates make LC3-II appears increased even when the level of interaction remains similar.

   To reply to this comment, we changed Figure 8. In Figure 8 of the revised manuscript we showed total PHB2 level as requested

3) It is very difficult to see if p62 level in PHB2 co-immunoprecipitates was less in mitochondrial fraction of BPV positive samples than in that of BPV negative samples in Figure 8 (indicated as figure 7) due to weak detection. In discussion, authors claimed that its significantly reduced (line 453 p17). More clear results need to be shown to support such discussion.

   To reply to this comment, we changed Figure 8. In Figure 8 of the revised manuscript we improved p62 expression levels as requested.

4) In addition to 3), is there any result to support that the recruitment of p62 is dependent on functional parkin in this article (line 353 p13). For example, perhaps inhibiting parkin?

   As we not perform any functional study, to avoid any possible confusion we removed: c) the recruitment of p62 may be dependent on functional parkin.

5) In figure 12, it seems less total TFEB in pathological samples than it is in healthy samples, yet its higher in pathological samples after fractionation shown in Figure 13. why?

   Actually, there was no significant differences in total TFEB in pathological samples vs healthy samples. We believe that mitochondrial and nuclear expression levels could be dependent on TFEB activity, that is transcriptionally activation of mitochondrial and nuclear target genes. Furthermore, very recently, TFEB was found to play a pivotal role in mitochondrial quality control (MQC) (Wang et al. Biomed Pharmacother. 128, 110272, 2020). It is worth noting that TFEB has been discovered quite recently (Settembre et al.- Science 332, 1429-1433, 2011). Therefore, many TFEB-modulated pathways are still to be elucidated (Pastore et al.  Nat Commun 11:2461, 2020).

6) As for ERAS interaction with PHB2 (line 435 p17), total PHB2 in the immunoprecipiates needs to be indicated. The same arguments here as 2) and 3),

  To reply to this comment, we changed Figure 8. In Figure 8 of the revised manuscript we showed total PHB2 level as requested.

7) Throughout the article, authors described that neoplastic samples are infected with BPV or harboring BPV. However, they only showed 3 samples express BPV E5 in figure S2. I believe not 100% of the disease is BPV positive. Thus, they should indicate how many samples are BPV positive and whether all “pathological sample” examined in this particle are BPV positive more clearly.

   All 15 “pathological samples” are É–PV positive as in all of them we detected expression of É–PV E5 oncoprotein by RT-PCR and/or by Western Blot. In line 208 we added: of all samples. In Figure 1 we showed three representative cases as in other published papers of ours we did. However, if the Reviewer wants to see a figure with all samples showing E5 oncoprotein expression, we can provide it to her/him.

   We have been studying for many years this spontaneous cattle disease in southern Italy believed to be an endemic area for this cattle pathology. Indeed temperate, humid climate typical of this area together acidity of pasturelands helps bracken fern to grow in overabundance. Cattle free at pasture in southern Italy usually graze on bracken-fern infested lands. Urinary bladder is the only target organ showing in vivo DNA adducts following to exposure to activated ptaquiloside (APT), a water soluble sesquiterpenoid glycoside of bracken fern. BPVs and APT act synergistically, resulting in bladder cancerogenesis. Outside endemic areas where bracken fern grows as well as in absence of BPVs, the prevalence of bladder cancer in cattle is much lower and basically negligible, thus representing 0.01% of all bovine malignancies seen in abattoirs (Meuten, Tumors in Domestic Anmals, Ames, Iowa: John Wiley and Sons Inc., 2017).

Minor points:

8) In line 343 p12 and line 350 p13, its figure 8 instead of figure 7.

We corrected these mistakes lines 348 and 354, respectively.

Round 3

Reviewer 2 Report

I am satisfied overall improvements that authors have made. However, some of the points that authors claimed to be modified are still in its original state. They need to be corrected to be acceptable for the publication.

(1) I am satisfied modification made in 336-338. I agree that results shown in this article are sufficient to state “ it is conceivable that PHB2 is a mitochondrial (line 338)” but NOT sufficient to state “confirm” (line 338). I have already indicate what kind of results needed to be included to sate “confirmation” in my previous latter.

(4) Authors indicated c) is removed. But still there

(7) I don’t doubt that authors have years of experience and great knowledge working on spontaneous cattle diseases. It may be common knowledge among veterinarians specializing cattle diseases that nearly 100% cases are caused by spontaneous BPV infection. On the other hand, “pathogens” targets readers in broader area of science including scientists not familiar with these facts. Therefore, I think it needs to be clearly stated as “all samples are BPV positive”.

Author Response

1) I am satisfied modification made in 336-338. I agree that results shown in this article are sufficient to state “ it is conceivable that PHB2 is a mitochondrial (line 338)” but NOT sufficient to state “confirm” (line 338). I have already indicate what kind of results needed to be included to sate “confirmation” in my previous latter.

As requested, we changed 'confirms' in 'suggests' (line 339)

(4) Authors indicated c) is removed. But still there

We removed the sentence (lines 358)

(7) I don’t doubt that authors have years of experience and great knowledge working on spontaneous cattle diseases. It may be common knowledge among veterinarians specializing cattle diseases that nearly 100% cases are caused by spontaneous BPV infection. On the other hand, “pathogens” targets readers in broader area of science including scientists not familiar with these facts. Therefore, I think it needs to be clearly stated as “all samples are BPV positive”.

To be clearer, we added "required to bladder carcinogenesis" (line 207).

This manuscript is a resubmission of an earlier submission. The following is a list of the peer review reports and author responses from that submission.

Round 1

Reviewer 1 Report

The study shows the role of PHB2 protein as a receptor involved in mitophagy in urothelial cells from cattle.

In Italy, Borzacchiello et al. (2003) detected the BPV-2 by a PCR assay in 76.7% (46/60) and 50% (17/34) of cattle urinary bladder samples with and without neoplastic lesions, respectively. Their control group consisted of urinary bladder samples without macroscopic lesions.

How was the BPV prevalence in group of fifteen 3- to 25-year-old cows in this study? Were all samples check by pathologist to prove the neoplastic lesion? How the samples were stored or preserved from degradation before the analysis?

Were, during sample processing, taken stringent precautions to avoid cross-contamination between samples?

The quality of pictures of western blots is very low. How some of the figures of poor quality could be analyzed by densitometry? No clear results as conclude by author are shown in figure 7.

Why are shown in all figures of western blots different numbers of healthy and pathological samples, none of figure showed all 25 samples together?

Some western blots showed different results of density of proteins in group of pathological samples (for example in figure 8 for OPTN protein), could authors comment this discrepant result of pathological samples? Is it the calculation of means optimal when it looks like there are two different group of pathological samples, one with high density of OPTN protein and one with comparable with control group?

In conclusion, could author summarize more the role of PHB2 protein and the potential role of BPV E5 in mitophagy?

Reviewer 2 Report

The manuscript by De Falco et al describes the involvement of Prohibitin 2 in parkin-mediated mitophagy in urothelium of cattle infected with BPV. While this would in principle constitute an interesting observation. Unfortunately, the experimental quality of the present work is not sufficient. In most figures, way too few samples have been analyzed for any generalization. Selection of individual sample sets for different experiments is not explained. The quality of the Western blots is in most cases not suitable for analysis for that matter. The immunoprecipitation experiments lack isotype controls and have been performed with too few samples for any conclusions. Their interpretation is further hampered by the low quality Western blot.

Major

The methods section states that there were 15 samples from diseased cows and 10 samples from healthy control animals.

In Figure 1, there are 3 diseased samples and 3 controls without further comment.

In Figure 3, there are 5 diseased samples and 3 controls.

In Figure 4, there are either 3 controls and 6 diseased samples, or 4 controls and 5 diseased samples, depending on how one interprets the labeling of the blots.

In Figure S2 it is 3 samples each.

In Figure 5, there are 4 healthy controls and 4 diseased samples.

In Figure 6, there are 3 healthy controls and 6 diseased samples.

In Figure 7, there are 2 controls and two samples.

In Figure 8, there are 4 controls and 4 samples.

In Figure 9, there are 4 controls and 4 samples.

In Figure 10, there are 3 healthy controls and 6 diseased samples.

This is a mess. I believe this to be accidental, but it somewhat does not exclude cherry picking of samples. Also, it is not clear to me on what basis samples were chosen for the individual analyses.

The sample set has not been consistently analyzed at all and it is not clear which samples were analyzed for which markers.

In Figure S3 there are – according to the legend - finally the 10 controls and 15 diseased samples, although it is completely unclear to me why the 15 samples are shown individually, and the controls are represented by one bar, and how this fits with the legend that states that the error bars represent SEM of three experiments in triplicates.

Figure 4.

This IP lacks an isotype control and is therefore not valid. Only a small fraction of the samples was analyzed, which precludes generalization. The quality of the Western blots is not acceptable for any publication and precludes any interpretation.

Figure 8.

There is an upregulation in two samples at most, the other two diseased samples don’t seem regulated. This dataset is too small to conclude anything.

Figure 8.

pTFEB may be downregulated in this sample, but again, this is only 4 samples out of 15

Figure 11

This IP lacks an isotype control and is therefore not valid. The small number of analyzed samples precludes generalization. The quality of the Western blots fpr pTFEB is not acceptable for publication and precludes interpretation.

Minor

Figure 2           I don’t see any arrows.

Reviewer 3 Report

       In this manuscript, the authors investigated the role of prohibition 2 (PHB2) in parkin-mediated mitophagy in urothelial cells infected with bovine papillomavirus (BPV). The authors observed the severe ultrastructural abnormalities of the inner mitochondrial membrane in the urothelial cells infected with BPV. Furthermore, they showed that the protein level of PHB2 in mitochondrial fraction is higher in the urothelial mucosa samples with BPV infection, compared with the healthy samples, indicating that PHB2 is associated with the observed abnormal mitochondrial distribution. In addition, the showed that the partners of PHBs including PINK1, parkin, LC2-II, TFEB, and P62 that cooperatively regulate the parkin-mediated mitophagy are enriched in mitochondrial factions of neoplastic urothelial mucosa samples. In general, the study is short of novelty, and the experiments are not well designed and performed. The comment are shown as below.

  1. The authors compared the expression of viral E5 mRNA, OPA1 protein, PHB2 protein, pTFEB protein and OPTN protein in urothelial samples collected from healthy cows and the cows clinically suffering from chronic enzootic haematuria. In each figure, the authors incorporated multiple samples for comparisons. The authors need to provide the detailed information of the samples used in each figure, especially the sample identification.
  2. Regarding cell fractionation and mitochondria isolation performed in this study, the level of marker protein for each fraction isolated from samples should be quantified as loading control for comparison. For instance, HSP90, a cytosol marker, LamA/C, a nucleus marker should be measured and quantified in the isolated fraction. They would not only serve as the indicators showing the purity of isolated fraction, but also allow us to use as a loading control to compare the level of interested targets in healthy and neoplastic urothelial mucosa samples.
  3. In Fig. 1, Please label the size of DNA ladder bands.
  4. In Fig. 2, Please include the TEM images of mitochondria from non-neoplastic urothelial mucosa samples for comparison. In addition, the indicated arrows are missing in Fig.2
  5. In line 249-255, the authors claimed that they could detect the PHB1 and PHB2 in the urothelial cells, however, there are no data present in the manuscript. Please include the data.
  6. In Fig.7, the authors need to clarify which line is corresponding to non-neoplastic sample, and which one is corresponding to neoplastic bladder sample.